# Morphology and size of the particles emitted from a gasoline direct injection-engine vehicle and their ageing in an environmental chamber

Jiaoping Xing[a,b], Longyi Shao[a] *, Wenbin Zhang[c], Jianfei Peng[d], Wenhua wang[a], Shijin Shuai[c], Min Hu[d], Daizhou Zhang[e] *

[a]State Key Laboratory of Coal Resources and Safe Mining, School of Geoscience and Survey Engineering, China University of Mining and Technology (Beijing), Beijing 100083, China.
[b]2011 Collaborative Innovation Center of Jiangxi Typical Trees Cultivation and Utilization, School of Forestry, Jiangxi Agricultural University, Nanchang, 330045, China.
[c]State Key Laboratory of Automotive Safety and Energy, Department of Automotive Engineering, Tsinghua University, Beijing 100084, China
[d]State Key Joint Laboratory of Environmental Simulation and Pollution Control, College of Environmental Sciences and Engineering, Peking University, Beijing 100871, China
[e]Faculty of Environmental and Symbiotic Sciences, Prefectural University of Kumamoto, Kumamoto 862-8502, Japan

* Corresponding Author - e-mail: shaoL@cumtb.edu.cn (Longyi Shao); dzzhang@pu-kumamoto.ac.jp (Daizhou Zhang)

## Highlights

1. GDI-engine vehicles emitted a large amount of both primary and secondary organic aerosols.

2. Higher contents of organic particles were emitted under hot stabilized running and hot start states.

3. Sulfate and secondary organic aerosol formed on the surface of primary particles after ageing.

4. Particles aged rapidly by catalyzed acidification under high pollution levels in Beijing.

## Abstract:

Air pollution is particularly severe in developing megacities, such as Beijing, where vehicles equipped with modern gasoline direct injection (GDI) engines are becoming one of major sources of the pollution. This study presents the characteristics of individual particles emitted by a GDI gasoline vehicle and their ageing in a smog chamber under the Beijing urban environment, as part of the Atmospheric Pollution & Human Health (APHH) research programme. Using transmission electron microscopy, we identified the particles emitted from a commercial GDI-engine vehicle running under various conditions, namely, cold start, hot start, hot stabilized running, idle, and acceleration states. Our results showed that most of the particles were organic, soot and Ca-rich ones, with small quantities of S-rich and metal-containing particles. In terms of particle size, the particles exhibited a bimodal distribution in number *vs* size, with one mode at 800–900 nm, and the other at 140–240 nm. The amounts of organic particles emitted under hot start and hot stabilized states were higher than those emitted under other conditions. The amount of soot particles was higher under cold start and acceleration states. Under the idle state, the proportion of Ca-rich particles was highest, although their absolute number was low. In addition to quantifying the types of particles emitted by the engine, we studied the ageing of the particles during 3.5 hours of photochemical oxidation in an environmental chamber under the Beijing urban environment. Ageing transformed soot particles into core-shell structures, coated by secondary organic species, while the content of sulfur in Ca-rich and organic particles increased. Overall, the majority of particles from GDI-engine vehicles were organic

and soot particles with submicron or nanometric size. The particles were highly reactive; they reacted in the atmosphere and changed their morphology and composition within hours via catalyzed acidification that involved gaseous pollutants at high pollution levels in Beijing.

## 1. Introduction

Air pollution caused by $PM_{2.5}$ in megacities such as Beijing, the capital city of China, is of public and academic concerns due to its environmental impacts (Bond et al., 2013; Huang et al., 2014; Liu et al., 2017) and adverse health effects (Chart-asa and Gibson, 2015; Shao et al., 2017). Motor vehicle emissions are one of the most significant sources of airborne particles in the urban atmosphere (Hwa and Yu, 2014), and contribute up to 31% of primary particulate emissions of $PM_{2.5}$ in Beijing (Yu et al., 2013). Moreover, secondary aerosol formation associated with traffic emissions is a major process leading to the rapid increase of $PM_{2.5}$, which results in severe haze episodes (Huang et al., 2014). Although emissions from gasoline engines are relatively lower than those from diesel engines (Alves et al., 2015), the number of gasoline-powered vehicles in urban areas greatly exceeds that of diesel-powered vehicles. The total number of vehicles in China reached 310 million in 2017, about 70% of these were powered by gasoline engines (National Bureau of Statistics of China, 2018). There are two main types of gasoline engines, namely, conventional multipoint port fuel injection (PFI) engines and gasoline direct injection (GDI) engines. In recent years, the demand for engines with high efficiency and low fuel consumption has led to an increasing use

of GDI engines in light-duty passenger cars. The market share of GDI-engine vehicles
has increased dramatically over the past decade and was estimated to reach 50% of new
gasoline vehicles sold in 2016 (Zimmerman et al., 2016). In Beijing and northern China,
the vehicle emissions become a more concerning issue in terms of air pollution when
the emission from coal combustion are seriously reduced after the Action for
Comprehensive Control of Air Pollution in Beijing since 2017 (Chang et al., 2019;
Chen et al. 2019; Zhang et al. 2019). In spite of this, regional transport of coal-burning
emissions from the surrounding areas can still influence the urban air sometimes
severely in winter (Ma et al., 2017; Zhang et al., 2019).

The number, mass and size distribution of particles emitted from GDI-engine

vehicles have been studied (Khalek et al., 2010; Maricq et al., 2011; Baral et al., 2011).
The size distribution usually has an accumulation mode with the maxima in the
diameter range of 100–300 nm. Major components of the particles include elemental
carbon (EC), organic carbon, and ash (Giechaskiel et al., 2014). Besides particulate
matter, the engines emit gaseous hydrocarbon compounds. These compounds might
form particles, or be adsorbed on the surface of particle aggregates, leading to the
growth of the particles in the engine emission (Luo et al., 2015). Relatively high particle
emissions by GDI-engine vehicles have prompted studies on the effects of engine
operating parameters and fuel composition on the characteristics of the particles (Hedge
et al., 2011; Szybist et al., 2011). It has been found that, in general, emissions under the
cold start condition make up the major contribution to the total amount of PM emissions
from GDI engines (Chen and Stone, 2011). Studies have also demonstrated that the
highest particle emissions from GDI engines in number concentration occur under the
acceleration state during transient vehicle operations (Chen et al., 2017).
Studies have also shown that gasoline vehicles are an important source of
secondary aerosol precursors in urban areas (Suarez-Bertoa et al., 2015). Secondary
aerosols can be formed via gas-phase reactions of volatile organic compounds and
multiphase and heterogeneous processes of primary particles (Zhu et al., 2017).
Experiments performed in environmental chambers demonstrated that the mass of
secondary aerosols derived from precursors could exceed that of directly-emitted
aerosols (Jathar et al., 2014). The occurrence of secondary aerosols on particles could
change the properties of particles in size, mass, chemical composition, morphology,
optical and hygroscopic parameters. These changes, in turn, might affect the
environmental impact of the particles significantly, for instance in terms of visibility,
human health, weather, and energy budgets (Laskin et al., 2015; Peng et al., 2017). In
general, the ageing processes of primary particles in the atmosphere are studied to
understand their climate effects (Niu et al., 2011). However, the lack of data on primary
particles emitted by gasoline engines hinders a deep understanding of the roles and
activities of the particles in ambient air pollution and relevant environmental effects.
Atmospheric Pollution & Human Health (APHH) research programme aimed to
explore the sources and processes affecting urban atmospheric pollution in Beijing.
Details regarding this project are given in Shi et al. (2018). To address one of the aims
of the AIRPOLL-Beijing (Source and Emissions of Air Pollutants) and AIRPRO-
Beijing (The integrated Study of AIR Pollution Processes), we employed a dedicated
experiment to investigate the characteristics of the individual particles, in terms of the
number concentration, size distribution, emitted from a GDI-engine vehicle during a
real-world driving cycle for chassis dynamometer test, i.e., the Beijing driving cycle
(BDC). Various test modes were introduced to accurately evaluate the emission from
light- or medium-duty vehicles. Furthermore, experiments were conducted in an
environmental chamber to investigate the ageing processes of particles emitted by GDI-
engine vehicles in ambient air in Beijing. We utilized a transmission electron
microscope equipped with an Oxford energy-dispersive X-ray spectrometer (TEM-
EDX) to identify the morphology, size and elemental composition of particles emitted
by the GDI-engine vehicle when it was running under different states. Particles before
and after a 3.5-hour ageing in the chamber were compared on the basis of the TEM-
EDX analysis. The TEM-EDX analysis provides the information on the internal
inhomogeneity, mixing state and surface characteristics of individual particles and has
been used to analyze the aerosol particles (Li and Shao, 2009; Loh et al., 2012; Adachi
and Buseck, 2015; Shao et al., 2017). The experimental design allows for the study of
the physical and chemical characteristics of the particles emitted from the GDI-engine
vehicles, as well as their ageing in a simulated urban atmosphere. The purposes of this
study are to evaluate the individual characteristics and the ageing process of primary
particles emitted by a GDI-engine vehicle, to investigate the ageing processes of such
particles in the atmosphere, and to deepen the understanding of the environmental
impact of gasoline-powered vehicle emissions.

## 2. Material and methods

### 2.1 Test vehicle, fuels, and test procedure

The GDI-engine vehicle utilized in the experiment complies with the China Phase 4 (equivalent to Euro 4) standard. It uses a three-way catalyst to reduce gaseous emissions. The GDI (model GDI-1.4-T) in the test vehicle is recognized as a representative of leading-edge designs of gasoline engines, because of its advanced engine technologies such as its better fuel burning efficiency and lower greenhouse gas emissions than other types of engine. Vehicles equipped with such GDI engines constitute the majority of light-duty vehicles in China, especially in large cities like Beijing. Details of the engine used in this study are listed in Table S1. The fuel used in the experiment is a commercial gasoline blend of common quality in China. The properties of the fuel were measured by SGS-CSTC Standards Technical Services Co., Ltd., China, and are listed in Table S2. The fuel has a Research Octane Number (RON) of 93 and is a fifth-stage gasoline. It contains 36.7% of aromatics and 15.4% of olefins in volume and has 6% of sulfur in mass, representing a typical fifth-stage gasoline in China (with high aromatics) and is now widely used in Beijing. The experiments were conducted within repeated Beijing Driving Cycles (BDCs), and one BCD included a 200-s "cold start" phase followed by an 867-s "hot stabilized running" phase. The conditions during a BDC in the experiments are illustrated in Figure S1a. The cold start state was achieved by starting the vehicle with a period of small accelerations, while the hot stabilized running state had multiple periods of large acceleration and a maximum velocity of 50 km h$^{-1}$. The BDC, characterized by a higher proportion of idling periods and a lower acceleration speed than the New European Driving Cycle

(NEDC), was performed to simulate the repeated braking and acceleration on road in
megacities such as Beijing.
All tests were performed on a Euro 5/LEV2/Tier 2-capable test cell on a 48-inch
single-roll chassis dynamometer at the State Key Laboratory of Automobile Safety and
Energy Conservation at Tsinghua University. The test procedure for each run was as
follows: fuel change, BDC preparation, soak, cold start BDC test, and hot start BDC
test. After fuel change and BDC preparation, the test vehicle was then conditioned with
an overnight soak for more than 10 h. The soak room temperature was maintained
between 20 and 30 °C. Due to the limitation of the facilities and available running time,
a hot start test was conducted within 5 mins after the cold start test. A dilution unit was
applied to dilute the exhaust from the tailpipe into 1/10 in volume using synthetic air
composed of 79% $N_2$ and 20% $O_2$, in order to obtain the concentrations suitable for
subsequent measurements and suppress possible coagulation. The number
concentration of the emitted particles was monitored by a Combustion Fast Particle
Size Spectrometer Differential Mobility Spectrometer 500 (DMS 500). The maximum
measurable number concentration of DMS 500 was $10^{11}$ (dN/dlogDp/cc) after the
dilution (Petzold et al., 2011). For the analyses of individual particles, 6–8 samples
were collected during one BDC test. At least one sample was collected under each
running state (i.e. cold start, hot start, idle state, acceleration state, or hot stabilized
running state). The driving cycle test was repeated at least twice. Two or more samples
were obtained for each running state. A single-stage cascade impactor (KB-2, Qingdao
Jinshida Company) was mounted to the exit of the tailpipe after the dilution unit. The
emitted particles were collected onto 300-mesh copper TEM grids, which were covered
with a carbon-coated formvar film. The flow rate was 1.0 L min$^{-1}$, and the cut-off
diameter of the impactor for 50% collection efficiency was 0.25 μm if the density of
the particles was 2 g cm$^{-3}$. For each sample, the collection time was 60 s.
**2.2 Environmental chamber experiments**

Particles from the GDI-engine vehicle were introduced into an environmental

chamber and exposed to sunlight. The chamber, made of perfluoroalkoxy (PFA) Teflon
in order to achieve a high transmission of ultraviolet light, has an internal volume of
1.2 m$^3$. Ambient sunlight was used as the driving force for photochemical reactions in
the chamber, in an environment close to actual open air. Before the experiments, the
chamber was cleaned by flushing with zero air for approximately 12 hours and
illuminated with sunlight, to remove residues that could influence the experiments.
$H_2O_2$ (1 mL, 30%), together with the vehicle emission, was injected into the chamber
to generate OH exposure. The OH exposure at the end of the experiments reproduced
extreme oxidation processes, which were equivalent to cases of an oxidation more than
10 days in Beijing ambient air if the 24-hour-mean concentration of OH is $10^6$
molecules cm$^{-3}$ (Lu et al., 2013). The aging experiments for the gasoline exhausts were
carried out with a relatively high OH exposure compared to ambient conditions in order
to obtain the aging process. This method and the amount of $H_2O_2$ have been frequently
used in smog chamber experiments (Song et al., 2007; Song et al., 2019). After the
injection, the experiments were conducted from approximately 13:00 to 17:00 local
time under sunshine, with the relative humidity being kept at around 50%. The global
solar radiation when the tests were carried out was approximately 318 W m$^{-2}$. After 3.5
h of ageing, the particles in the chamber were collected onto mesh TEM grids using the
impactor. The collection time for each sample was 120 s. The schematic diagram of the
experimental system is presented in Figure S1b.

**2.3 TEM/EDX and scanning transmission electron microscopy (STEM) analyses**
The particles in the samples were examined using a Tecnai G2 F30 field emission
high-resolution transmission electron microscope (FE-HRTEM). This microscope is
also equipped with an Oxford EDX and a STEM unit with a high-angle annular dark-
field detector (HAADF). The EDX can detect elements with the atom number larger
than 5 (B) in a single particle. The HAADF can detect the distribution of a certain
element by mapping the distribution of the element in a particle. The TEM was operated
with the acceleration voltage of 300 kV. EDX spectra were firstly collected for 20 live
seconds to minimize the influence of radiation exposure and potential beam damage
and then for 90 live seconds for a range of possible elements. Copper was excluded
from the analysis because of interference from the TEM grids which were made of
copper.
To ensure the representativeness of the analyzed particles, more than 150 particles
from at least 3 random areas were analyzed from the center and periphery of the
sampling spot on each grid. All individual particles larger than 50 nm in the selected
areas were analyzed. The TEM images were digitized using an automated fringe image
processing system named Microscopic Particle Size of Digital Image Analysis System
(UK) to project the surface areas of the particles. The equivalent spherical diameter of
a particle was calculated from its projected area, expressed as the square root of 4 A/π,

where A was the projected area. The electron microscope analysis of individual particles

was very time consuming, which hindered us from analyzing more particles from

multiple engines emission. There are differences in emissions from vehicles to vehicles,

even for vehicles with same model engines. Only one GDI vehicle, the type of which

constitutes the majority of light-duty vehicles in China, was tested in this study. The

representativeness of the present results remains unevaluated carefully with, such as,

comparisons between vehicles to achieve broader statistical results, although the tests

in the present studies were conducted under strict control conditions.

## 3. Results

### 3.1. Particle morphology, elemental composition and size

A total of 2880 particles were analyzed from the GDI-engine vehicles. Most of the

particles were in the sub-micrometer size range. Based on morphology and elemental

composition of the particles, the majority of them were identified as soot, organic and

Ca-rich particles, a smaller amount was identified as S-rich or metal-rich particles (Fig.

1). The method of particle classification is similar to that adopted by Okada et al. (2005)

and Xing et al. (2019). In the following description, "X-rich" means that the element

"X" occupies the largest proportion in the element composition of the particles. Figure

2 illustrates the number-size distributions of the relative concentration (dN/dlogD) of

primary particles from the GDI-engine vehicle, where N is the relative number fraction

and D is the equivalent diameter. The particles were in the range of 60–2500 nm and

displayed a bimodal distribution, with one mode in the 140–240 nm range, and another

in the 800–900 nm range. Particles smaller than 250 nm were largely underestimated

because of the loss during the particle collection. Therefore, there should have been

more particles in the smaller mode range than shown in Figure 2. Concerning the loss
of small particles, we measured the size distribution by the DMS500 (Fig. S2). The
results showed that a large amount of nucleation mode particles were emitted by the
GDI vehicle.
It should be noted that organic particles were mainly composed of C and O
elements, and contained a small amount of inorganic elements Ca, P, S and Zn.
Elemental mapping of the organic particles exhibited the presence of Ca, P, S and Zn in
some of the particles, showing the mixture state of organic and inorganic materials (Fig.
1f). It has been reported that such particles could be related to the combustion of fuels
or lubrication oil (Rönkkö et al., 2013). In addition to these primary organic particles,
the GDI-engine vehicle emitted precursor gases, which produced secondary organic
particles via gas-phase reactions, and multiphase and heterogeneous processes on the
primary particles. A group of spherical particles were found in the environmental
chamber (Fig. 1g). These particles became semi-transparent or transparent to an
electron beam, which was characteristic of organic materials, liquid water, or their
evaporation residues either mixed or not mixed with electron absorptive materials. We
regarded these particles as secondary organic particles because the humidity in the
chamber during the experiment was kept much below saturation (relative humidity
around 50%). Therefore, these particles were expected to mainly consist of secondary
organic materials, which should have been produced via gas phase reactions or on the
surface of pre-existing particles (Hu et al., 2016). No other elements, except C and O,
were identified in these particles, which was consistent with the above inference.
Similar particles were also encountered in other environmental chamber experiments
studying emissions from light-duty gasoline vehicles (Jathar et al., 2014).

**3.2 Number fractions of particles**
Figure 3 illustrates the numbers of accumulation mode particles emitted by
burning one kilogram of fuel during the cold start and hot start driving cycles. PM
emissions at the start-up stage under both cold and hot start states were higher than the
emissions under the states when the engine was fully warmed and the vehicle operation
was stabilized. The PM emission was the highest under the hot stabilized running state
($2.3\times10^{10}$ particles (kg fuel)$^{-1}$), followed by those under the hot start ($1.2\times10^{10}$ particles
(kg fuel)$^{-1}$), cold start ($7.1\times10^{9}$ particles (kg fuel)$^{-1}$), and acceleration running states
($2.9\times10^{9}$ particles (kg fuel)$^{-1}$), with the emission under the idle running state being the
lowest ($7.4\times10^{8}$ particles (kg fuel)$^{-1}$) (Fig. S3). The higher emission of particle in term
of number for the GDI vehicle under the hot start state can be ascribed to the
experimental time of the vehicle engine. The hot start test in this study was conducted
within 5 mins after the cold start test. The PM emissions from GDI vehicles were
relatively less affected by ambient temperature for the initial 30 minutes during the
warming up of the engines (Cotte et al., 2001). This may lead to the high value of the
PM emission for the hot start state which is slightly higher than that for the cold start
state. Although the total PM emission were higher under hot start state than that under
the cold start state, the comparison of those in the size range of accumulation mode
indicates that the particulate emissions for this mode of particles were higher under the
cold start state than under the hot start sate (Fig. 3). This can be attributed to the less
efficiency of the vaporization of fuel droplets in the combustion cylinder under the cold
start state (Chen et al., 2017). Size distributions of the particles varied with driving
conditions (Fig. S4). Under the cold start state and acceleration running state, higher
number concentrations, and thus higher mass concentrations of the particles with
accumulation mode were emitted in comparison with other running states.
Under all the running states, morphologies and types of the particles remained
similar but the proportions of different types of particles differed considerably (Fig. S5).
The proportion of organic particles was high under hot stabilized and hot start states.
Soot particles were abundant under cold start and acceleration states. A relatively higher
proportion of Ca-rich particles was found under idle state, compared to those under
other running states.
We estimated the number of different type particles in the emission under the
running states by burning one kilogram of fuel (Fig. 4). Organic particles in the
emission under the hot stabilized running state ($2.3 \times 10^9$ particles (kg fuel)$^{-1}$) and the
hot start running state ($3.6 \times 10^8$ particles (kg fuel)$^{-1}$) were higher than in the emission
under other running states. The number of soot particles were higher under the hot
stabilized running state ($1.7 \times 10^9$ particles (kg fuel)$^{-1}$) and the cold start state ($5.9 \times 10^8$
particles (kg fuel)$^{-1}$) than those under other running states. Under the idle state, the
relative proportion of Ca-rich particles was the highest, although their absolute number
was low ($1.4 \times 10^9$ particles (kg fuel)$^{-1}$).
Under the cold start state, a significant proportion of the emitted particles were
soot particles. This can be attributed to the incomplete vaporization of fuel droplets in
the combustion cylinder (Chen et al., 2017). Under the hot start state and the hot
stabilized running state, organic particles were predominant. Under these two running
states, the engine temperature was high, which enabled the fuel to evaporate and mix
with the air easily. With the increase of the temperature in the cylinders, the rate of
particle oxidation increased, which could cause an increase of organic particles in the
emission (Fu et al., 2014). Under the idle state, the fuel consumption was much lower
than that under the other running states, which resulted in a higher relative contribution
to particles from lubricant oil. The high Ca content in the lubricant oil led to a higher
Ca-rich particle emission under this running state. Under the acceleration state, the
predominant types of particle included soot, organic, and Ca-rich particles. As the
acceleration running required a high vehicular speed and engine load, the emissions
contained more soot particles than those under other running states.
**3.3. Aged particles in the environmental chamber**

A large amount of secondary organic particles (accounting for 80%-85% in

number), some soot particles, Ca-rich particles, and primary organic particles were
detected in the environmental chamber (Fig. 5). After the ageing process, many soot
particles changed into core-shell structures and became coated with secondary species
(Figs. 5b and 5c). The EDX results showed that almost all coatings were mainly
composed of C, O, and S, suggesting these coatings were a mixture of organic and
sulfate. The morphology and compositions of Ca-rich particles and organic particles
(Figs. 5e and 5g) changed, with the aged ones having a more irregular shape and a
higher sulfur content in comparison with fresh ones (Figs. 5A and B).

Approximately 80% of the soot particles were present in core-shell structures and

coated with secondary species after the 3.5-hour ageing. In contrast, before the ageing, the particles with a core-shell structure were only about 10% of the total. The mean diameter of the soot particles after ageing was around 0.49 μm, which was much smaller than that before the ageing (0.65 μm), indicating the shrinkage of the soot particles during the ageing (Fig. 5b). The core-shell ratios, defined as the ratio of the diameter of the core part (Dcore) to the diameter of the whole particle (Dshell) (Niu et al., 2016; Hou et al., 2018), were used to quantify the aging degree of the soot particles with coating. It was found that the core-shell ratios of the soot particles in the smog chamber were mainly in the range of 0.25–0.78, indicating the stronger aging degree of soot particles in the chamber than case data in urban air with the ratios of 0.4–0.9 (Niu et al., 2016)."

## 4. Discussion

### 4.1. Contribution of GDI-engine vehicle emissions to urban air pollution

Our investigation showed that the GDI-engine vehicle emitted a large amount of organic particles (32%), soot (32%), Ca-rich particles (26%), S-rich (5%) and metal-containing particles (4%). Relevant studies have also shown that the primary carbonaceous aerosols (element carbon + primary organic aerosol) accounted for 85 % of the PM in the GDI vehicles, suggesting that carbonaceous aerosols were the major contributors in the PM from GDI gasoline vehicles (Du et al., 2018). Considering the large fraction of the vehicles equipped with GDI engines in megacities like Beijing, this indicates a possible substantial contribution of GDI-engine vehicles to urban air pollution. Moreover, organic particles occupied the majority of the particles emitted

under hot stabilized running and hot start states. It has been noted that the organic matter
was the major component of the total particle mass during the hot start conditions
(Fushimi et al., 2016; Chen et al., 2017), which was consistent with the results obtained
for the number concentrations in our study. The hot stabilized running state is the most
frequent running condition of vehicles, whereas the hot start state is the most frequent
condition in congested traffic. This suggests that a substantial number of organic
compounds in the air pollution of populated cities might be directly related to vehicle
emissions.

Organic particles and soot particles in ambient air are emitted from a range of

sources including fossil fuels, biomass burning and urban waste burning (Kanakidou et
al., 2005). Table 1 shows the major characteristics of particles in the emissions from
different sources. For instance, there is a higher fraction of soot particles and a lower
fraction of organic particles in the emissions of GDI-engine vehicles compared to PFI-
engine vehicles (Xing et al., 2017). Organic particles in emissions from gasoline
vehicles are usually enriched in Ca, S and P (Xing et al., 2017; Liati et al., 2018). In
comparison, emissions from biomass/wood burning are usually dominated by organic
particles, which account for more than 50% of the total amount of particles (Liu et al.,
2017). Furthermore, organic particles from biomass/wood burning usually show
elevated K content, and thus, this element is frequently used as an indicator for
biomass/wood burning organic particles (Niu et al., 2016). Observations of primary
particles directly from coal burning have also demonstrated a predominance of organic
particles, soot particles, S-rich particles and mineral particles (Zhang et al., 2018; Wang
et al., 2019). Both biomass burning and coal combustion can produce organic particles
and almost all the emitted particles contain a certain amount of Si in addition to C and
O. Table 1 also shows the elemental concentrations in the organic particles in the
emissions from different types of sources. Since the concentrations of minor elements
in the organic particles are highly dependent on the sources, they could be used for
source identification of individual particles in the atmosphere.

The present data also permit the compilation of a rough inventory of particle

categories and amounts emitted from GDI-engine vehicles under various running
conditions (Fig. 4). Combined with statistics on the number of vehicles with GDI
engines, the running time and the running conditions on roads within a certain area, it
is possible to make an approximate estimate of the amounts of primary particles emitted
from GDI-engine vehicles. Such estimate is the basis for accurate source apportionment
of particles from vehicles, and it will be very beneficial for studies on the anthropogenic
sources of primary particles in urban air. These data could be brought together to better
understand the sources of air pollutants in the Beijing megacity and to improve the
capability of developing cost-effective mitigation measures.
**4.2 Rapid ageing of primary particles in Beijing**

The results of chamber experiments indicate that sulfate and secondary organic

aerosol (SOA) form on the surface of soot, Ca-rich and organic particles. Moreover, the
atmospheric transformation of primary particles emitted by the GDI-engine vehicles
could occur within 3.5 hours, indicating the ageing was rapid. Peng et al. (2014) found
similar timescales for black carbon transformation under polluted conditions in Beijing.
The rapid ageing of primary particles could be caused by several factors, such as the
concentration of gaseous pollutants from the vehicles, strength of solar radiation,
relative humidity (RH), and $O_3$ concentration (Guo et al., 2012; Deng et al., 2017; Du
et al., 2018). The present experiments were conducted in the atmosphere with relative
humidity of approximately 50% and solar radiation of 318 $Wm^{-2}$. The total hydrocarbon
emission (THC) from the GDI vehicles was 0.297 g $km^{-1}$. Repeated braking and
acceleration in the BDC could cause incomplete combustion and consequently high
THC emission. Under a high concentration of gaseous pollutants, primary particles
would age rapidly when exposed to solar radiation. Consequently, secondary species
including SOA and sulfate were produced on or condensed onto the particles, leading
to the coating. Guo et al. (2014) also showed that secondary photochemical growth of
fine aerosols during the initial stage of haze development could be attributed to highly
elevated levels of gaseous pollutants.
The mixture of SOA and sulfate have been detected in our chamber experiment,
indicating the involvement of inorganic salts in the SOA formation. Previous studies
have demonstrated the enhancement of SOA production in the presence of inorganic
sulfate (Beardsley and Jang, 2015; Kuwata et al., 2015), and this is because sulfate can
catalyze carbonyl heterogeneous reactions, and consequently, lead to SOA production
(Jang et al., 2002; Jang et al., 2004). Moreover, these aged primary particles favored
the formation of secondary aerosols by providing reaction sites and reaction catalysts.
Sulfate and secondary organic aerosol (SOA) co-existed on the surface of primary
particles, such as soot, Ca-rich and organic particles. In addition, the products of VOCs
oxidation could react with $SO_2$ to rapidly produce sulfate (Mauldin et al., 2012). Thus,
the rapid ageing of primary particles could also be attributable to the acid-catalyzed
mechanism. As the major source of pollutants in urban air, the GDI-engine vehicles
supply both primary particles and precursor gaseous species, and the rapid ageing of
the particles under certain conditions is very likely to be the major driving force for the
elevation of urban air pollution.
**4.3 Implications and perspectives**
Our results indicated that GDI-engine vehicles emitted a large amount of both
primary and secondary organic aerosols. PM number emission of organic particles from
GDI-engine vehicle were $2.9 \times 10^9$ particles (kg fuel)$^{-1}$ during the BDC. Secondary
organic particle was predominant in the secondary aerosols, accounting for 80-85%%
particles in the chamber. Organic aerosols (OA) play an important role in the Earth's
radiation balance not only for its absorption and scattering of solar radiation but also
because they can alter the microphysical properties of clouds (Scott et al., 2014).
Particle size, shape, mixing state and composition affect their light scatterings and
absorption cross sections, and cloud condensation nuclei activity (Jacobson, 2001). OA
are composed of various types of chemical compounds with varying absorption
properties (mixing state), which are determined by the emission sources, the formation
mechanism (Zhu et al., 2017), and the source regions (Laskin et al., 2015). Primary OA
from biomass burning is co-emitted with soot (black carbon), inorganic salts, and fly
ash, producing internally and externally mixed particles in which the organic
components are present in different relative abundance (Lack et al., 2012). Similarity,
primary OA in the exhaust of gasoline and diesel vehicles are mixed with Ca, P, Mg,
Zn, Fe, S, and minor Sn inorganic compounds (Liati et al., 2018). In addition, previous
measurements have indicated that SOA usually exists as an internal mixture with other
aerosols, such as sulfate, ammonium, or nitrate (Zhu et al., 2017). Our results showed
that the POA emitted from GDI-engine vehicle were mixed with soot, inorganic
components such as Ca, P, and Zn. Some of the SOA formed in the smog chamber were
mixed with sulfate. The complexity of mixing state makes it difficult to characterize the
properties of OA. Lang-Yona et al. (2010) have found that for aerosols consisting of a
strongly absorbing core coated by a non-absorbing shell, the Mie theory prediction
deviate from the measurements by up to 10%. Moreover, atmospheric aging process,
involving aqueous-phase aging and atmospheric oxidation, can either enhance or
reduce light absorption by OA (Bones et al., 2010). The condensation process may
result in a dramatic enhancement of hydrolysis of OA compounds, affecting their
absorption spectra (Lambe et al., 2015).

Our results also showed that primary organic aerosols (POA) emitted by GDI-

engine vehicles could acquire OA and sulfate coatings rapidly, within a few hours, and
increase a sizable fraction of total ambient aerosols existing as internal mixtures. In
addition, the fast ageing further caused the increase of aged POA in the total OA,
consequently, largely modified the properties of the particles such as their optical
properties. The results of the experiments in the chamber showed that most of the aged
POA had a core-shell structure, whereas most of the secondary organic aerosols (SOA)
produced by gas-phase reactions had a uniform structure. These results push forward
the understanding on the mixing state and chemical composition of both POA and SOA.
The experimental data will benefit the parameterization of vehicles emissions in
numerical models dealing with urban air pollution.
**5. Conclusions**

1.  Five types of individual particles emitted by a GDI-engine gasoline vehicle were
identified, including soot, organic, Ca-rich, S-rich, and metal-rich particles. Among
them, soot, organic, and Ca-rich particles were predominant. The particles emitted
from this commercial GDI-engine gasoline vehicle displayed a bimodal size
distribution.
2.  The concentrations of the particles emitted by this commercial GDI-engine
gasoline vehicle vary with different running conditions. The PM emission was the
highest under the hot stabilized running state, followed by those under the hot start,
cold start, and acceleration running states, with the emission under the idle running
state being the lowest under the idle running state.
3.  The relative proportions of the different types of particles emitted by this
commercial GDI-engine gasoline vehicle varied with different running conditions.
Large amounts of organic particles were emitted during hot stabilized and hot start
states. Under cold start and acceleration states, the emissions were enriched in soot
particles. Under idle state, a relatively higher number of Ca-rich particles was
emitted, although the absolute number was low.
4.  After ageing in the environmental chamber, the structure of the soot particles
changed into a core-shell structure, and the particles were coated with condensed
secondary organic material. Ca-rich particles and organic particles also were
modified, and their content of sulfur increased after ageing.
5. Ageing of the emitted particles occurred rapidly, within hours. Such rapid ageing
could be attributable to an acid-catalyzed mechanism and to the high initial
concentrations of gaseous pollutants emitted by this commercial GDI-engine
gasoline vehicle.

## Data availability

All data presented in this paper are available upon request. Please contact the
corresponding author (shaoL@cumtb.edu.cn).

## Author contribution

LS designed this study; JX performed the experiments. JX, LS, DZ summarized
the data and wrote the paper. WZ, JP, WW, SS, MH supported the experiments and
commented the paper.

## Competing interests

The authors declare that they have no conflict of interest.

## Acknowledgements

This work was supported by Projects of International Cooperation and Exchanges
NSFC (Grant No. 41571130031). The data analysis was partly supported by Science
and Technology Project Founded by the Education Department of Jiangxi Province (No.
GJJ180226), Yue Qi Scholar Fund of China University of Mining and Technology
(Beijing), and a Grant-in-Aid for Scientific Research (B) (No.16H02942) from the
JSPS.

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

| Study | Source | Particle of type and relative percentages | Chemical composition of organic particles |
|---|---|---|---|
| This study | GDI-engine vehicles | Organic particles (OM) (32%), Soot (32%), Ca-rich particles (26%), S-rich (5%) and metal-containing particles (4%) | OM with Ca and weak P, S, and Zn |
| Xing et al. (2019) | PFI-engine vehicles | OM (44%), soot (23%), Ca-rich particles (20%), S-rich (6%) and metal-containing particles (6%). | OM with Ca and weak P, S, and Zn. |
| Liati et al. (2018) | GDI, PFI and diesel vehicles | Soot, OM (called ash-bearing soot particles) and ash particles. | OM with Ca, S, P, Fe and minor Zn. |
| Liu et al. (2017) | Crop residue combustion | OM (27%), OM-K (43%), OM-soot-K (27%), soot-OM (3%). | OM particles with K content. |
| Liu et al. (2017) | Wood combustion | OM (16%), soot (18%), OM-K (22%), OM-soot-K (15%), soot-OM (29%). | OM particles with K content. |
| Wang et al. (2019) | Coal burning | OM (38%), soot (40%), S-rich particles (2%), and mineral particles (18%). | OM mainly consisted of C, O and Si. |
| Zhang et al. (2018) | Residential coal burning | OM (51%), OM-S (24%), soot-OM (23%), S-rich (1%), metal-rich particles (1%), mineral particles (1%). | OM contained C, O, and Si with minor amounts of S and Cl. |



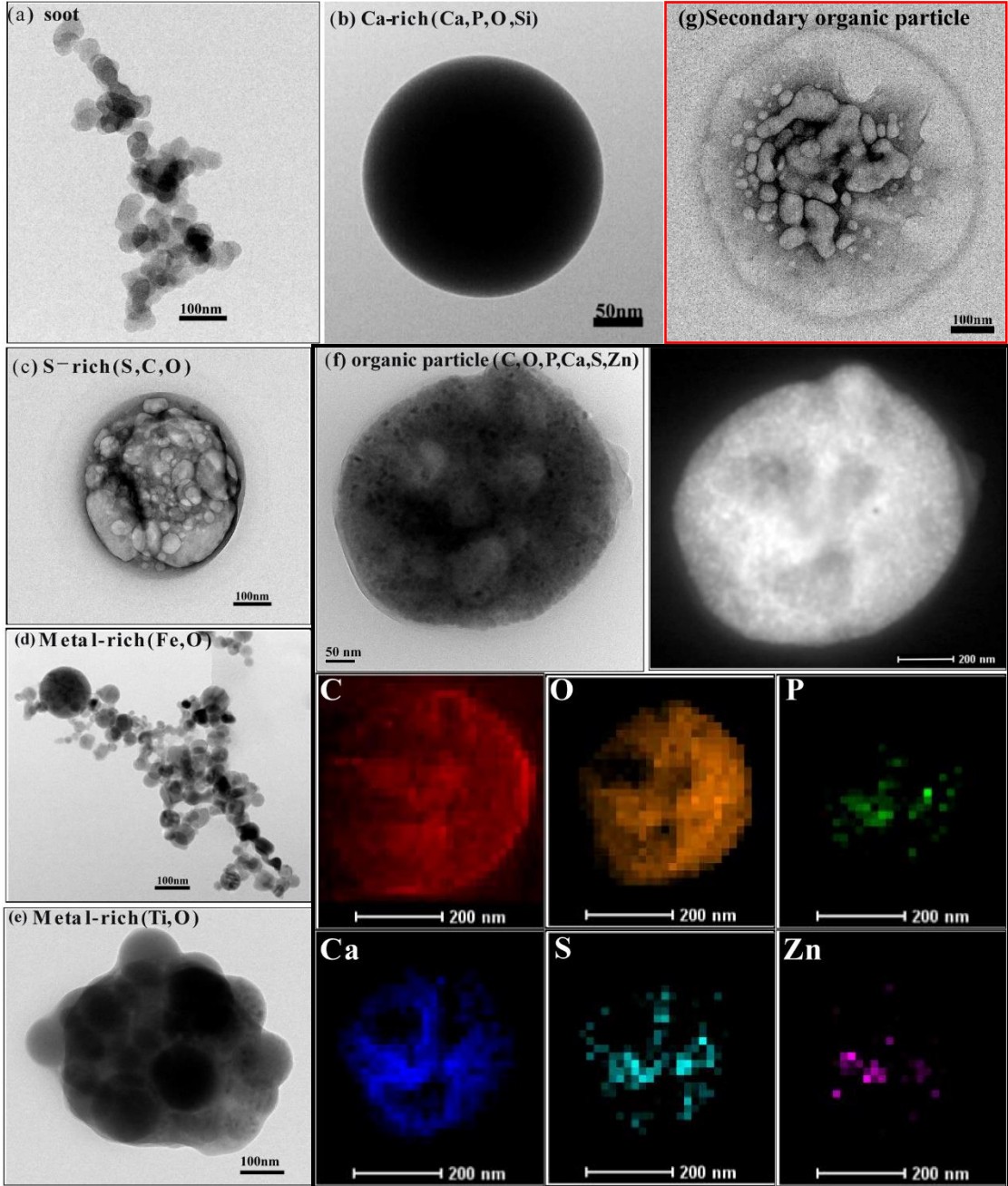

Figure 1. TEM images of the individual primary particles emitted from the GDI-engine
gasoline vehicle and the secondary organic particle in the chamber after exposure to
ambient sunlight for 3.5 hours. (a) soot particle; (b) Ca-rich particle; (c) S-rich particles;
(d) Metal-rich particles (Fe); (e) Metal-rich particles (Ti); (f) bright-field-TEM and
dark-field-TEM image of organic particles, and others are the mapping of the C, O, P,
Ca, S, and Zn in the organic particle; (g) secondary organic particle in chamber.

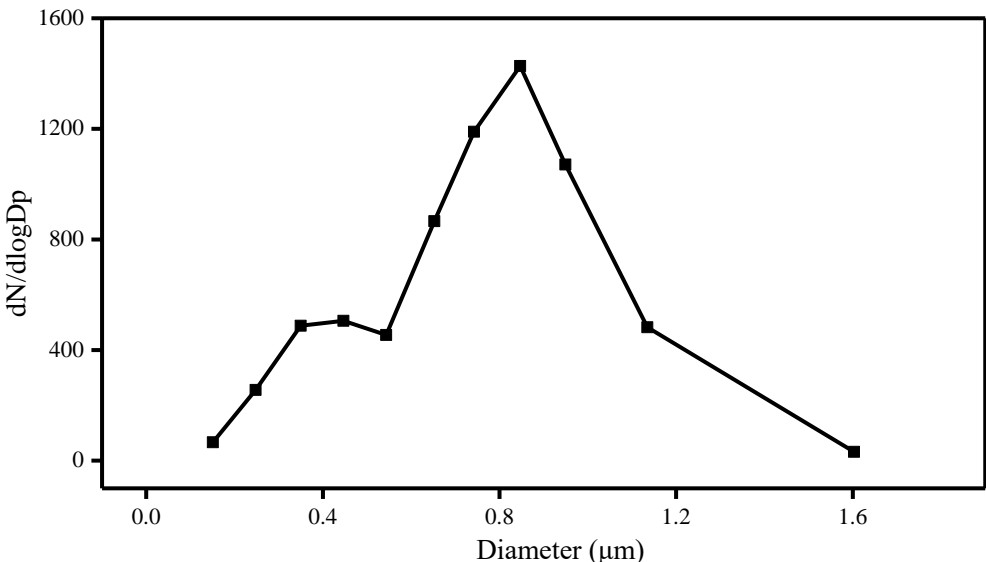


Figure 2. Size distribution of analyzed particles emitted from the GDI-engine gasoline
vehicles by the TEM images. In total, 2880 particles were analyzed from the GDI-
engine vehicles. Particles smaller than 0.25 μm should have been underestimated
because of the collection efficiency of the impactor.


(a)

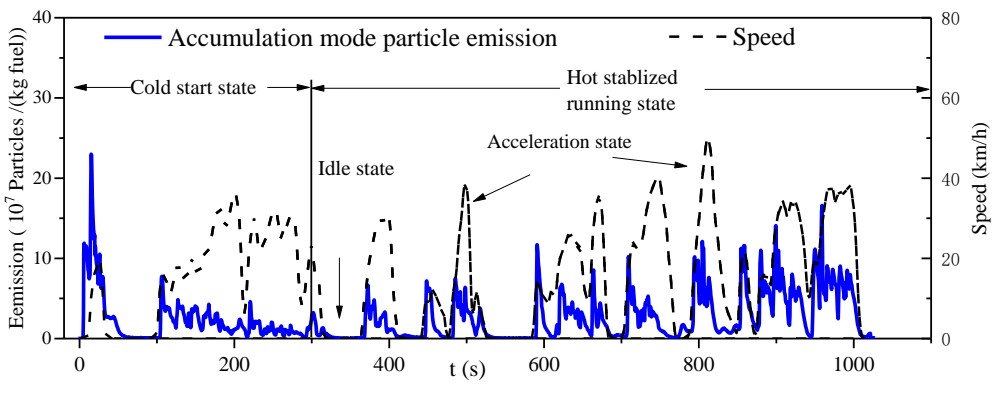


(b)

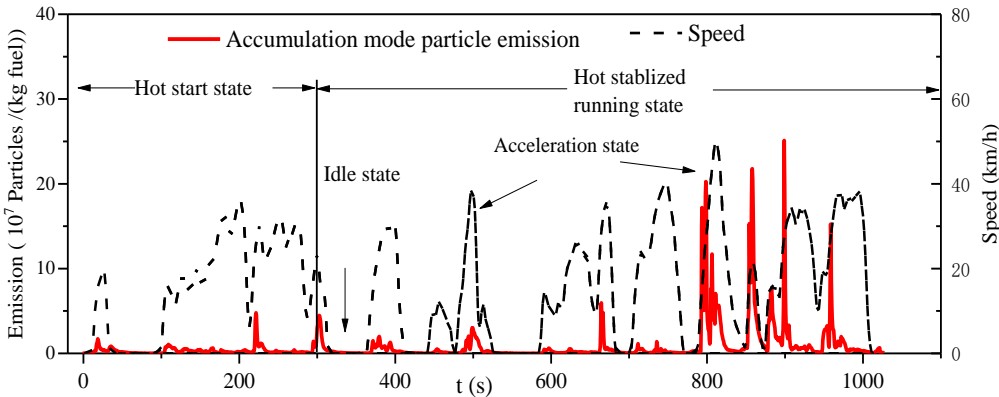


Figure 3. Particles in accumulation mode from the GDI vehicle during cold start (a) and
hot start (b) driving cycle. The vehicle speed is also shown for reference. Before the
test with cold start, the temperatures of the engine coolant and oil could not differ by
more than 2 °C during the soak temperature. The hot start test was conducted within 5
mins after the cold start test. The number concentration of particles during the tests was
monitored by DMS 500.



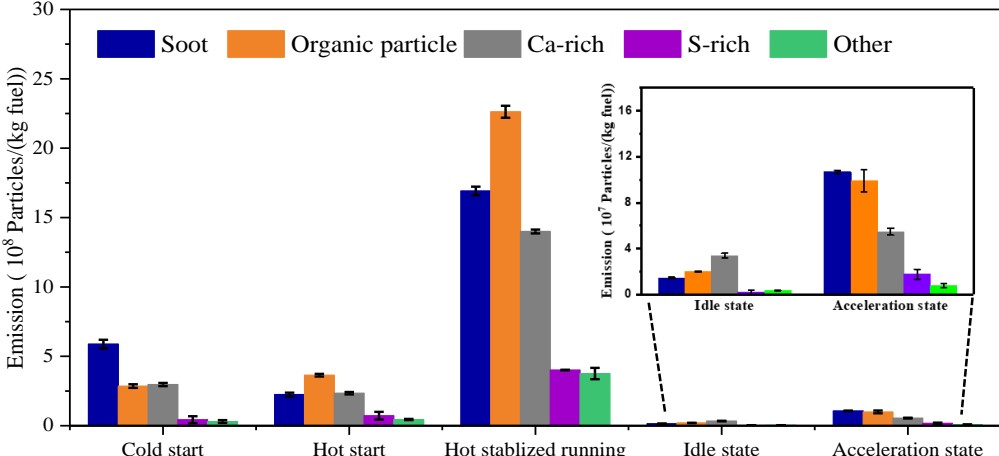


Figure 4. The number of different type particles in the emissions from the GDI vehicle under the different running states by the burning of per unit of fuel, including cold start, hot start, hot stabilization, idle, and acceleration states. Data presented as mean ± standard deviation, N = 3.


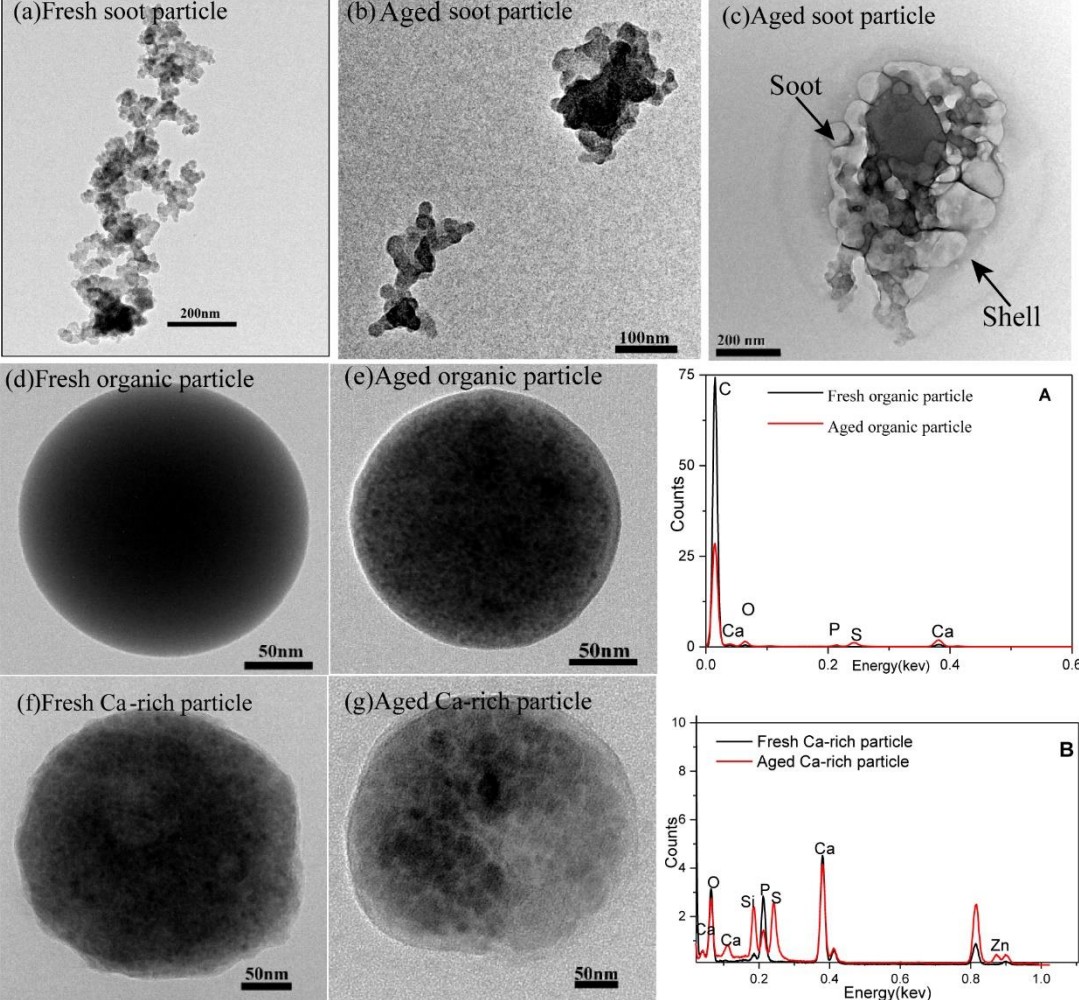


Figure 5. TEM images of particles in the chamber after exposure to ambient sunlight
for 3.5 hours. (a) Fresh soot particles; (b) Aged soot particles; (c) Aged soot particle (d)
Fresh organic particle (e) Aged organic particle (f) Fresh Ca-rich particle (g) Aged Ca-
rich particle (A) EDX spectrum for a fresh organic particle and an aged organic particle.
(B) EDX spectrum for a fresh organic particle and an aged Ca-rich particle.