# Peer review of "Morphology and size of the particles emitted from a gasoline"

_Atmospheric Chemistry and Physics, 2019_

## Referee Comment (RC1) · Anonymous Referee #1 · 28 Sep 2019

It is a well designed and prepared work related with the emission of single particle emission from gasoline vehicle emission and aging activities, with methods including TEM-EDX, chassis dynamometer, smoke chamber in ambient air condition, etc. adopted. It should be emphasized that this kind of works are still limited, especially in China, as the limitation of equipments, sampling platform, and so on. I really admire the study design. The data also provide evidence of the impact of gasoline vehicle emission on ambient air of megacities like Beijing. I suggested it can be accepted for ACP. Some concerns are listed below. 1. Line 80-83, coal combustion is still a problem in northern China. The authors should clarify this sentence. 2. I am not sure why H2O2 was added in the chamber and why such amounts were added. What is the concentrations

for the formation of OH through H2O2 photolysis. More detailed information should be listed. 4. Can you get the real solar radiation data from local weather bureau. 5. Why acid-catalyzed mechanism important for SOA formation in vehicle emission aging? Do you have other data or deep analysis?

———————————————

---

## Referee Comment (RC2) · Anonymous Referee #2 · 3 Oct 2019

The paper discusses the composition, size and mixing of single particles emitted by a gasoline direct injection (GDI) vehicle as determined using single particle electron microscopy and spectroscopy. I found the topic very interesting and important. I think the overall data and analysis is of interest and sufficient to warrant publication; however, I believe the paper could improve substantially with some changes and maybe a few relatively minor additional analyses that could be performed with the data available. Before the paper is published, the authors need to address and discuss several items as mentioned in the following general and specific comments.

General comments: - If I understood correctly the conclusions of this study are based

on only one vehicle. While I totally understand that single particle analyses are very time consuming and analyzing emissions from several engines would be prohibitive, it is well known that there is tremendous vehicle to vehicle emission variability, even for the same model and engine (several works published in the literature are available on the topic). So my request here is not that to add data, which would be well beyond the scope of the study, but to clearly mention this caveat and limitation and discuss briefly its potential implications. Related to that, is the vehicle representative of the average GDI vehicles in Beijing? If so how? If not, how can the conclusions of the paper be generalized as the authors attempt to claim? (E.g., In the method section, they mention that the vehicle represents a "leading-edge design", does that means that most of the other GDI engine would perform worse than that in terms of emissions?)

- The paper is quite biased and limited in terms of citations of existing literature (some example will be discussed in the specific comments but more pervade the paper) and the paper would be much more impactful if put in perspective of a large body of existing literature for example on PM vehicle emission (not only in China but also in other countries), single particle analysis, particle optical properties measurements, and effects of single particle mixing geometries on calculated or measured optical properties and radiative effects, etc.

- Overall the paper is quite clearly written, but some additional English grammar checks would improve readability; this should include reducing typos and checks for tense consistency.

- The atmospheric implications section is too generic and not always substantiated by the results or the provided citations (more on this in the specific comments below). This section should be made more concrete and provide a deeper and more significant discussion.

Specific comments: - Title: It is a matter of personal taste, but I typically prefer not to use acronyms in the title, so the authors could consider spelling out GDI to target a

wider audience.

- Highlights: "Particles from a GDI-engine vehicle and their ageing were studied." To me this is not a highlight, it is just the topic of the paper.

- Line 38: "... must be paid enough attention." Something seems missing here and the sentence reads awkward. Consider rephrasing it.

- Line 45: "grain size" does not seem to be a very common term in the community... maybe "particle size" or just "size"?

- Lines 65 – 66: circular sentence, as it is now it reads as if particulate matter PM is a source of airborne particles. The authors instead mean vehicles are a source of PM, I think. Consider rephrasing.

- Line 81: change "concerned" with "concerning"

- Line 82: "compressed" maybe should be "reduced"?

- Line 130: "could provide" should be "provide" or "provided"

- Line 153: This is mostly a curiosity for me, but it seems like 6% of sulfur in the fuel is quite high. Is that the norm? Are those fractions by weight or by mass?

- Line 159: A max speed of 50 kmh-1 seems a bit low, even if that might be the speed limit in the city, do real vehicle actually respect that limit? Can the author discuss this point?

- Line 178: Please provide the manufacturer and model of the impactor.

- Line 215: Please provide more information on the image analysis procedure, including, if available, citations to existing literature, software or methods.

- Figure 2 reports the number size distribution of the particles emitted. I believe that's from the electron microscopy, that should be made clear. Additionally, what engine state is the distribution representing, or is it a composite of all the particles collected,

maybe this was mentioned somewhere, but I could not find that information. Related to it, on line 230 the authors mentioned a range of 60 to 2500 nm, but earlier on, when describing the impactor, they mention a lower 50% size-cut of 250 nm, so the 250 to 60 nm contribution is probably severely underestimated, therefore the detailed of the first mode (140-240 nm) is probably severely biased by the sampling. Related to this, I think there would be a lot of benefit to either add a plot in figure 2 or present a separate plot with the size distribution that should be available from the DMS500 instrument mentioned in line 172.

- Line 260: It is interesting that the particle concentration was higher for hot conditions that cold conditions. This sparks the question though if the size distribution of the particles also changed and with it the mass emission... again using the DMS500 data should easily answer that question at least for the ensemble particle size distributions.

- Line 272: "of" missing after between "type" and "particles"

- The sentence from line 288 to 290. This is an example (there are other instances throughout the paper) of verb tense inconsistency.

- Line 299: It would be nice to know if there is a semi-quantitative assessment of what the coating material was for the most part. Sulfates? Organics? Others...

- Line 300: "...organic particles changed..." what kind of changes, please elaborate?

- Line 314: "...the majority..." in number, but is that the case in mass as well, please discuss.

- Lines 350-351: It would be useful to provide information on what is the most common coating material and, if possible, provide an estimate of the core to shell ratio. That ratio can be key to optical calculations to understand the impact of absorption increase due to coating and mentioned later in the paper. A quantitative determination of the core to shell ratio for soot, that might be possible to be determined from the data available to the authors, could be very useful to the community and make the paper substantially

more impactful.

- Line 356: How high were the gas concentrations? Were they representative of re-alistic atmospheric conditions? If not, could one extrapolate on what would be an equivalent aging time in more realistic atmospheric concentrations? Some comments on this issue are needed.

- Line 360: the sentence sparks the question, was there any contribution from break and tire tear emissions? Were those captured? Probably not because, if I recall cor-rectly, these samples were collected on an engine dynamometer, but it might be good to comment (one can use several results from previous studies available in the litera-ture) on potential contributions of tailpipe emissions vs. break and tires emissions.

- Line 375: The acid-catalyzed mechanism is mentioned here and even mentioned as one of the highlights of the paper, but the discussion is minimal, or inexistent here. This can be an interesting and important point, so please provide some elaboration of this topic and provide some unbiased citations of relevant literature on the topic.

- Line 380: The large contribution of GDI emissions here is argued, but there is no data collected or model performed (even if just conceptual) to really quantify, at least semi-quantitatively, what the contribution could be with respect to other sources. The only discussion I recall is on the elemental composition regarding tracers to distinguish different sources, but no discussion to quantify their potential contribution (also in what terms? Mass? Number? Something else?). So the "considerable potential contribu-tion" statement is a vague ill-posed guess that is not really proven here as the data are presented. One could try to estimate the contribution by calculating for example estimated total contribution of GDI engines (by multiplying the fuel-based emission fac-tor by the fraction of fuel consumed by GDI vehicles in the area) vs. the total particle burden in the region... or some other estimate exercise of this sort. Otherwise the sentence cannot be supported by the evidence provided.

- Lines 386-388: This is a clear example, among others, of biased representation (or

lack of) previous work in this paper. There have been very many previous (definitely previous to 2017) studies showing that individual particles, including organics come internally mixed with other particles, so this statement is not very true (especially in terms of "recent") and biased in terms of literature discussion and citations.

- Lines 391-393: Again limited (poor) literature citation choice and a bit of simplistic view of the effect of internal mixing on aerosol radiation interaction (which I think is what the authors refer to here). Please improve the discussion and support it with some more reprehensive and balanced citations judiciously chosen from the large body of previous literature in the field. Another such example appears in lines 399 to 400.

- Lines 403 to 404: In this case, I would say the sentence is flatly untrue; there are plenty of studies on the optical properties of POA, and the authors should discuss them and cite them accordingly.

- Figure 2: As mentioned earlier, specify that this distribution is from the electron microscopy analysis, and overlap or add a side plot with the size distribution from the DMS500.

- Figure 4: For regulatory applications, it could be interesting to generate a similar figure but for estimated mass (or at least volume) fractions.

[Figure]

---

## Author Comment (AC1) · 5 Dec 2019

Point-to-point responses to the comments:

Anonymous Referee #1 It is a well designed and prepared work related with the emission of single particle emission from gasoline vehicle emission and aging activities, with methods including TEMEDX, chassis dynamometer, smoke chamber in ambient air condition, etc. adopted. It should be emphasized that this kind of works are still limited, especially in China, as the limitation of equipments, sampling platform, and so on. I really admire the study design. The data also provide evidence of the impact of gasoline vehicle emission on ambient air of megacities like Beijing. I suggested it can

be accepted for ACP. Some concerns are listed below. Response: The authors show their appreciation to the reviewer for the comments and suggestions. Here I describe the point-to-point responses to the comments and questions.

1. Line 80-83, coal combustion is still a problem in northern China. The authors should clarify this sentence. Response: We fully agree with this comment. In order to avoid the misunderstanding, we added "In spite of this, regional transport of coal-burning emissions from the surrounding areas can still influence the urban air sometimes severely in winter (Ma et al., 2017; Zhang et al., 2019)" in the end of the mentioned paragraph.

References: Ma, Q., Wu, Y., Zhang, D., Wang, X., Xia, Y., Liu, X., Tian, P., Han, Z., Xia, X., Wang, Y., and Zhang, R.: Roles of regional transport and heterogeneous reactions in the PM2.5 increase during winter haze episodes in Beijing, Sci Total Environ, 599, 246-253, 10.1016/j.scitotenv.2017.04.193, 2017. Zhang, M., Li, Z., Xu, M., Yue, J., Cai, Z., Yung, K.K.L., and Li, R.: Pollution characteristics, source apportionment and health risks assessment of fine particulate matter during a typical winter and summer time period in urban Taiyuan, China, Hum Ecol Risk Assess, 10.1080/10807039.2019.1684184, 2019.

2. I am not sure why H2O2 was added in the chamber and why such amounts were added. What is the concentrations for the formation of OH through H2O2 photolysis. More detailed information should be listed. Response: H2O2 was injected into the chamber as the source of hydroxyl radical (OH). Please refer to descriptions in line 206-212. In the revision, to make this more clear, we have added in line 206-212 "Assuming the 24 hr mean concentration of 106 OH molecules cm-3 in Beijing (Lu et al., 2013), the OH exposure at the end of the experiments reproduced extreme oxidation processes in the atmosphere, which is equivalent to cases of an oxidation more than 10 days. The aging experiments for the gasoline exhausts were carried out with a relatively high OH exposure compared to ambient conditions in order to obtain the aging process. This method and the amount of H2O2 have been frequently used in smog chamber experiments (Song et al., 2007; Song et al., 2019)."

References: Lu, K.D., Hofzumahaus, A., Holland, F., Bohn, B., Brauers, T., Fuchs, H., Hu, M., Häseler, R., Kita, K., Kondo, Y., Li, X., Lou, S.R., Oebel, A., Shao, M., Zeng, L.M., Wahner, A., Zhu, T., Zhang, Y.H., and Rohrer, F.: Missing OH source in a suburban environment near Beijing: observed and modelled OH and HO2 concentrations in summer 2006, Atmos Chem Phys, 13, 1057-1080, 10.5194/acp-13-1057-2013, 2013. Song, C., Na, K., Warren, B., Malloy, Q., and Cocker, D.R.: Secondary Organic Aerosol Formation from m-Xylene in the Absence of NOx, Environ Sci Technol, 41, 7409-7416, 10.1021/es070429r, 2007. Song, M., Zhang, C., Wu, H., Mu, Y., Ma, Z., Zhang, Y., Liu, J., and Li, X.: The influence of OH concentration on SOA formation from isoprene photooxidation, Sci Total Environ, 650, 951-957, 10.1016/j.scitotenv.2018.09.084, 2019.

3. Can you get the real solar radiation data from local weather bureau. Response: We are not able to obtain the in-situ solar radiation data from local meteorology bureau. For this reason, we use the simulated data.

4. Why acid-catalyzed mechanism important for SOA formation in vehicle emission aging? Do you have other data or deep analysis? Response: According to published literature (Kuwata et al., 2015; Jang et al., 2002; Jang et al., 2004; Beardsley and Jang, 2015), we believe that the acid-catalyzed mechanisms are major path for SOA formation, and at least our results are consistent with the results.

In the revision, the following descriptions are added in line 416-429. "The mixture of SOA and sulfate have been detected in our chamber experiment, indicating the involvement of inorganic salts in the SOA formation. Previous studies have demonstrated the enhancement of SOA production in the presence of inorganic sulfate (Beardsley and Jang, 2015; Kuwata et al., 2015), and this is because sulfate can catalyze carbonyl heterogeneous reactions, and consequently, lead to SOA production (Jang et al., 2002; Jang et al., 2004). Sulfate and secondary organic aerosol (SOA) co-existed on the surface of primary particles, such as soot, Ca-rich and organic particles. In addition, the products of VOCs oxidation could react with SO2 to rapidly produce sulfate (Mauldin et al., 2012)." All these results have demonstrated the potential importance of

acid-catalyzed mechanisms for SOA formation in aging of vehicle emission. The references: Beardsley, R.L., and Jang, M.: Simulating the SOA formation of isoprene from partitioning and aerosol phase reactions in the presence of inorganics, Atmospheric Chemistry and Physics Discussions, 15, 33121-33159, 10.5194/acpd-15-33121-2015, 2015. Kuwata, M., Liu, Y., McKinney, K., and Martin, S.T.: Physical state and acidity of inorganic sulfate can regulate the production of secondary organic material from isoprene photooxidation products, Physical chemistry chemical physics: PCCP, 17, 5670-5678, 10.1039/C4CP04942J, 2015.
* * *

---

## Author Comment (AC2) · 5 Dec 2019

Point-to-point responses to the comments: Anonymous Referee #2 The paper discusses the composition, size and mixing of single particles emitted by a gasoline direct injection (GDI) vehicle as determined using single particle electron microscopy and spectroscopy. I found the topic very interesting and important. I think the overall data and analysis is of interest and sufficient to warrant publication; however, I believe the paper could improve substantially with some changes and maybe a few relatively minor additional analyses that could be performed with the data available. Before the paper is published, the authors need to address and discuss several items as mentioned in
the following general and specific comments.

Response: The authors show their appreciation to the reviewer for the comments and suggestions. Here I describe the point-to-point responses to the comments and questions.

General comments: 1. If I understood correctly the conclusions of this study are based on only one vehicle. While I totally understand that single particle analyses are very time consuming and analyzing emissions from several engines would be prohibitive, it is well known that there is tremendous vehicle to vehicle emission variability, even for the same model and engine (several works published in the literature are available on the topic). So my request here is not that to add data, which would be well beyond the scope of the study, but to clearly mention this caveat and limitation and discuss briefly its potential implications. Related to that, is the vehicle representative of the average GDI vehicles in Beijing? If so how? If not, how can the conclusions of the paper be generalized as the authors attempt to claim? (E.g., In the method section, they mention that the vehicle represents a "leading-edge design", does that means that most of the other GDI engine would perform worse than that in terms of emissions?)

Response: We fully agree with the comments on the limitation of the results from the present experiment with one engine. In the revision, the following descriptions are added in line 238-240. "The electron microscope analysis of individual particles is very time consuming, which hindered us from analyzing more particles from multiple engines emission." To the question on the meaning of "leading-edge design", the answer is yes. This type of GDI engines perform better than other type engines such as the traditional port fuel injection (PFI) engines for higher fuel burning efficiency, lower greenhouse gas emissions, and better fuel economy.

In the revision, we have rephrased the sentence "The GDI (model GDI-1.4-T) in the test vehicle is recognized as a representative of leading-edge designs of gasoline engines, having advanced engine technologies, that combine turbocharging and GDI together

with a downsized displacement." into "The GDI (model GDI-1.4-T) in the test vehicle is recognized as a representative of leading-edge designs of gasoline engines, because of its advanced engine technologies such as its better fuel burning efficiency and lower greenhouse gas emissions than other types of engine" (line 149-154 in the revision).

2. The paper is quite biased and limited in terms of citations of existing literature (some example will be discussed in the specific comments but more pervade the paper) and the paper would be much more impactful if put in perspective of a large body of existing literature for example on PM vehicle emission (not only in China but also in other countries), single particle analysis, particle optical properties measurements, and effects of single particle mixing geometries on calculated or measured optical properties and radiative effects, etc.

Response: We try to add more discussions on the points raised in this comment, but feel that such discussions made the manuscript tedious and give an impression of the exaggeration of present results because of the limited data. However, we do have added four important references, which are dealing with the points of PM vehicle emission (Hwa and Yu, 2014), single particle analysis (Loh et al., 2012), particle optical properties measurements(Laskin et al., 2015), effects of single particle mixing geometries on calculated or measured optical properties and radiative effects (Lack et al., 2012) in the parts of relevant discussion (Line 69, Line139, Line 450, Line 452).

The references: Hwa, M., and Yu, T.: Development of real-world driving cycles and estimation of emission factors for in-use light-duty gasoline vehicles in urban areas, Environ Monit Assess, 186, 3985-3994, 10.1007/s10661-014-3673-1, 2014. Lack, D.A., Langridge, J.M., Bahreini, R., Cappa, C.D., Middlebrook, A.M., and Schwarz, J.P.: Brown carbon and internal mixing in biomass burning particles, Proceedings of the National Academy of Sciences, 109, 14802-14807, 10.1073/pnas.1206575109, 2012. Laskin, A., Laskin, J., and Nizkorodov, S.A.: Chemistry of Atmospheric Brown Carbon, Chem Rev, 115, 4335-4382, 10.1021/cr5006167, 2015. Loh, N.D., Hampton, C.Y., Martin, A.V., Starodub, D., Sierra, R.G., Barty, A., Aquila, A., Schulz, J., Lomb, L.,

Steinbrener, J., Shoeman, R.L., Kassemeyer, S., Bostedt, C., Bozek, J., Epp, S.W., Erk, B., Hartmann, R., Rolles, D., Rudenko, A., Rudek, B., Foucar, L., Kimmel, N., Weidenspointner, G., Hauser, G., Holl, P., Pedersoli, E., Liang, M., Hunter, M.M., Gumprecht, L., Coppola, N., Wunderer, C., Graafsma, H., Maia, F.R.N.C., Ekeberg, T., Hantke, M., Fleckenstein, H., Hirsemann, H., Nass, K., White, T.A., Tobias, H.J., Farquar, G.R., Benner, W.H., Hau-Riege, S.P., Reich, C., Hartmann, A., Soltau, H., Marchesini, S., Bajt, S., Barthelmess, M., Bucksbaum, P., Hodgson, K.O., Strueder, L., Ullrich, J., Frank, M., Schlichting, I., Chapman, H.N., and Bogan, M.J.: Fractal morphology, imaging and mass spectrometry of single aerosol particles in flight, Nature, 486, 513-517, 10.1038/nature11222, 2012.

3. Overall the paper is quite clearly written, but some additional English grammar checks would improve readability; this should include reducing typos and checks for tense consistency.

Response: We have reduced typos and checked for tense consistency. The final version of the re-written manuscript was edited by a professional Elsevier Language Editing Services.

4. The atmospheric implications section is too generic and not always substantiated by the results or the provided citations (more on this in the specific comments below). This section should be made more concrete and provide a deeper and more significant discussion.

Response: We have tried our best to make this part more concrete and accurate.

In the revision, a semi-quantitative assessment of the different particles emitted by the GDI-engine vehicle has been addressed, and the following descriptions are added in line 352. "Our investigation shows that the GDI-engine vehicle emitted a large amount of organic particles (32%), soot (32%), and Ca-rich particles (26%), S-rich (5%) and metal-containing particles (4%)".
In line 438-441, we have added "PM number emission of organic particles from the GDI-engine vehicle were 2.9×109 particles (kg fuel)-1 during the BDC. Secondary organic particles were predominant in the secondary aerosols, accounting for 80-85% particles in the chamber."

Also, in line 342-348, an estimate of the core to shell ratio for soot particles and the atmospheric implication were discussed and the following sentence is provided, "The core-shell ratios, defined as the ratio of the diameter of the core part (Dcore) to the diameter of the whole particle (Dshell) (Niu et al., 2016; Hou et al., 2018), were used to quantify the aging degree of the soot particles with coating. It was found that the core-shell ratios of the soot particles in the smog chamber were mainly in the range of 0.25–0.78, indicating the stronger aging degree of soot particles in the chamber than case data in urban air with the ratios of 0.4–0.9 (Niu et al., 2016)."

In line 469-473, we removed "However, studies on the optical properties of OA have been mainly focused on SOA, and only a few studies dealt with POA. Our results indicate the possible substantial contribution of emissions from the GDI-engine vehicles to POA, especially in traffic congestion. For a better understanding of the roles that traffic emissions play in urban air pollution, further segregation of the aerosol particles such as POA and SOA in model and observation studies is inevitable."

The references: Hou, C., Shao, L., Hu, W., Zhang, D., Zhao, C., Xing, J., Huang, X., and Hu, M.: Characteristics and aging of traffic-derived particles in a highway tunnel at a coastal city in southern China, Sci Total Environ, 619-620, 1385-1393, 10.1016/j.scitotenv.2017.11.165, 2018. Niu, H., Hu, W., Zhang, D., Wu, Z., Guo, S., Pian, W., Cheng, W., and Hu, M.: Variations of fine particle physiochemical properties during a heavy haze episode in the winter of Beijing, Sci Total Environ, 571, 103-109, 10.1016/j.scitotenv.2016.07.147, 2016.

Specific comments:

5. Title: It is a matter of personal taste, but I typically prefer not to use acronyms in the

title, so the authors could consider spelling out GDI to target a wider audience.

Response: GDI in the title is spell out in the revision i.e. gasoline direct injection.

6. Highlights: "Particles from a GDI-engine vehicle and their ageing were studied." To me this is not a highlight, it is just the topic of the paper.

Response: "Particles from a GDI-engine vehicle and their ageing were studied" was removed from the Highlights.

7. Line 38: "...must be paid enough attention." Something seems missing here and the sentence reads awkward. Consider rephrasing it.

Response: We rephrased "... where pollutants from vehicles equipped with modern gasoline direct injection (GDI) engines must be paid enough attention." as "…..... where vehicles equipped with modern gasoline direct injection (GDI) engines are becoming one of major sources of the pollution".

8. Line 45: "grain size" does not seem to be a very common term in the community. . . maybe "particle size" or just "size"?

Response: We rephrased "grain size" as "particle size".

9. Lines 65 – 66: circular sentence, as it is now it reads as if particulate matter PM is a source of airborne particles. The authors instead mean vehicles are a source of PM, I think. Consider rephrasing.

Response: We rephrased it as "Motor vehicles emissions are one of the most significant sources of airborne particles in the urban atmosphere".

10. Line 81: change "concerned" with "concerning"

Response: We changed.

11. Line 82: "compressed" maybe should be "reduced"?

Response: "compressed" was replaced with "reduced".

12. Line 130: "could provide" should be "provide" or "provided"

Response: We removed "could".

13. Line 153: This is mostly a curiosity for me, but it seems like 6% of sulfur in the fuel is quite high. Is that the norm? Are those fractions by weight or by mass?

Response: The fuel contained 36.7% aromatics (by volume), 15.4% olefins (by volume), and 7% sulphur (by mass). It is the fifth-stage gasoline in China (with high aromatics) and is now widely used in Beijing. Line 162-164, we revised "It contains, in volume, 36.7% of aromatics and 15.4% of olefins; it also has 6% of sulfur in mass." as "It contains 36.7% of aromatics and 15.4% of olefins in volume and has 6% of sulfur in mass, representing a typical fifth-stage gasoline in China (with high aromatics) and is now widely used in Beijing."

14. Line 159: A max speed of 50 km h-1 seems a bit low, even if that might be the speed limit in the city, do real vehicle actually respect that limit? Can the author discuss this point?

Response: The BDC, characterized by a higher proportion of idling periods and a lower acceleration speed than the New European Driving Cycle (NEDC), was performed to simulate the repeated braking and acceleration on road in megacities such as Beijing. The BDC was lasting about 17 min, with the highest speed being about 50 km h-1 (Fig. S1a). On the point whether the maximum speed is high or low, we have no data to show a further discussion. In fact, in Beijing, the real average speed is not fast because of frequent traffic gam. In the revision, the following descriptions are added in line 170-173. "The BDC, characterized by a higher proportion of idling periods and a lower acceleration speed than the New European Driving Cycle (NEDC), was performed to simulate the repeated braking and acceleration on road in megacities such as Beijing."

15. Line 178: Please provide the manufacturer and model of the impactor.

Response: The manufacturer and model of the impactor (KB-2, Qingdao Jinshida

Company) was added in line 192-193.

16. Line 215: Please provide more information on the image analysis procedure, including, if available, citations to existing literature, software or methods.

Response: We added "named Microscopic Particle Size of Digital Image Analysis System (UK)." in line 235-236.

17. Figure 2 reports the number size distribution of the particles emitted. I believe that's from the electron microscopy, that should be made clear. Additionally, what engine state is the distribution representing, or is it a composite of all the particles collected, maybe this was mentioned somewhere, but I could not find that information. Related to it, on line 230 the authors mentioned a range of 60 to 2500 nm, but earlier on, when describing the impactor, they mention a lower 50% size-cut of 250 nm, so the 250 to 60 nm contribution is probably severely underestimated, therefore the detailed of the first mode (140-240 nm) is probably severely biased by the sampling. Related to this, I think there would be a lot of benefit to either add a plot in figure 2 or present a separate plot with the size distribution that should be available from the DMS500 instrument mentioned in line 172.

Response: For the caption of Figure 2, we added "...by TEM images...". On the comments about particles in the range smaller than 250 nm, we added the size distribution by the DMS500 instrument as Figure S2. In line 256-259, we added that "Concerning the loss of small particles, we measured the size distribution by the DMS500 (Figure S2). The results showed that a large amount of nucleation mode particles were emitted by the GDI vehicle."

18. Line 260: It is interesting that the particle concentration was higher for hot conditions that cold conditions. This sparks the question though if the size distribution of the particles also changed and with it the mass emission... again using the DMS500 data should easily answer that question at least for the ensemble particle size distributions.

Response: We have added the size distribution during the Beijing driving cycle by the DMS500 instrument as Figure S4. In line 290-294, we have added that "Size distributions of the particles varied with driving conditions (Figure S4). Under the cold start state and acceleration running state, higher number concentrations, and thus higher mass concentrations of the particles with accumulation mode were emitted in comparison with other running states."

19. Line 272: "of" missing after between "type" and "particles"

Response: "of" was added.

20. The sentence from line 288 to 290. This is an example (there are other instances throughout the paper) of verb tense inconsistency.

Response: We have modified the verb tense to match the previous example. In line 317-319, we changed "Under the idle state, the fuel consumption was much lower than that under the other running states, which results in the relative contribution of lubricant oil to particles in the emission being higher" as "Under the idle state, the fuel consumption was much lower than that under the other running states, which resulted in a higher relative contribution to particles from lubricant oil". The whole text was revised for this issue.

21. Line 299: It would be nice to know if there is a semi-quantitative assessment of what the coating material was for the most part. Sulfates? Organics? Others...

Response: Our results indicated that almost all the coatings were a mixture of organic and sulfate. The EDX results showed that the coating was mainly composed of C, O, and S. In line 331-333, we have added that "The EDX results showed that almost all coatings were mainly composed of C, O, and S, suggesting these coatings were a mixture of organic and sulfate."

22. Line 300: ". . .organic particles changed. . ." what kind of changes, please elaborate?

Response: In line 334-336, We rephrased "the aged ones had a higher sulfur (S) content in comparison with the fresh ones (Figs. 5A and B)"as "with the aged ones having a more irregular shape and a higher sulfur content in comparison with fresh ones (Figs. 5A and B)."

23. Line 314: ". . .the majority. . ." in number, but is that the case in mass as well, please discuss.

Response: The single particle analysis is a good approach for studying particles in number, but not in mass. So, the discussion was focused on particles in number. We tried to discuss the mass by citing relevant publications. We added that "It has been noted that the organic matter was the major component of the total particle mass during the hot start conditions (Chen et al., 2017; Fushimi et al., 2016), which is consistent with the results obtained for the number concentrations in our study" (line 360-363 in the revision). The references: Chen, L., Liang, Z., Zhang, X., and Shuai, S.: Characterizing particulate matter emissions from GDI and PFI vehicles under transient and cold start conditions, Fuel, 189, 131-140, 10.1016/j.fuel.2016.10.055, 2017. Fushimi, A., Kondo, Y., Kobayashi, S., Fujitani, Y., Saitoh, K., Takami, A., and Tanabe, K.: Chemical composition and source of fine and nanoparticles from recent direct injection gasoline passenger cars: Effects of fuel and ambient temperature, Atmos Environ, 124, 77-84, 10.1016/j.atmosenv.2015.11.017, 2016.

24. Lines 350-351: It would be useful to provide information on what is the most common coating material and, if possible, provide an estimate of the core to shell ratio. That ratio can be key to optical calculations to understand the impact of absorption increase due to coating and mentioned later in the paper. A quantitative determination of the core to shell ratio for soot, that might be possible to be determined from the data available to the authors, could be very useful to the community and make the paper substantially more impactful.

Response: In line 331-333, we have added "The EDX results showed almost all the

coatings was mainly composed of C, O, and S, suggesting these coatings were a mixture of organic and sulfate. In line342-348, we have added "The core-shell ratios, defined as the ratio of the diameter of the core part (Dcore) to the diameter of the whole particle (Dshell) ( Niu et al., 2016; Hou et al., 2018), were used to quantify the aging degree of the soot particles with coating. It was found that the core-shell ratios of the soot particles in the smog chamber were mainly in the range of 0.25–0.78, indicating the stronger aging degree of soot particles in the chamber than case data in urban air with the ratios of 0.4–0.9 (Niu et al., 2016)." The references: Hou, C., Shao, L., Hu, W., Zhang, D., Zhao, C., Xing, J., Huang, X., and Hu, M.: Characteristics and aging of traffic-derived particles in a highway tunnel at a coastal city in southern China, Sci Total Environ, 619-620, 1385-1393, 10.1016/j.scitotenv.2017.11.165, 2018. Niu, H., Hu, W., Zhang, D., Wu, Z., Guo, S., Pian, W., Cheng, W., and Hu, M.: Variations of fine particle physiochemical properties during a heavy haze episode in the winter of Beijing, Sci Total Environ, 571, 103-109, 10.1016/j.scitotenv.2016.07.147, 2016.

25. Line 356: How high were the gas concentrations? Were they representative of realistic atmospheric conditions? If not, could one extrapolate on what would be an equivalent aging time in more realistic atmospheric concentrations? Some comments on this issue are needed.

Response: In this study, we have tried to obtain the data on aging of the primary particles emitted by the GDI vehicle, with high initial gaseous pollutant concentrations and a strong oxidization environment in the chamber. The total hydrocarbon emission (THC) from the GDI vehicles was 0.297 g km-1. The OH exposure at the end of the experiments reproduced extreme oxidation processes in the atmosphere, which is equivalent to cases of an oxidation more than 10 days.

In Line 206-212, We have added "Assuming the 24 hr mean concentration of 106 OH molecules cm-3 in Beijing (Lu et al., 2013), the OH exposure at the end of the experiments reproduced extreme oxidation processes in the atmosphere, which is equivalent to cases of an oxidation more than 10 days. The aging experiments for the gasoline

exhausts were carried out with a relatively high OH exposure compared to ambient conditions in order to obtain the aging process."

26. Line 360: the sentence sparks the question, was there any contribution from break and tire tear emissions? Were those captured? Probably not because, if I recall correctly, these samples were collected on an engine dynamometer, but it might be good to comment (one can use several results from previous studies available in the literature) on potential contributions of tailpipe emissions vs. break and tires emissions.

Response: PM emitted from traffic were often derived from exhaust emissions (tailpipe exhaust emissions from gasoline and diesel engines) and non-exhaust emissions (wear debris from brake and tires, and road dust churned up by vehicle fleets). As the reviewer mentioned, our samples were collected on an engine dynamometer. So these samples were derived from exhaust emissions, and there was not any contribution from break and tire tear emissions.

27. Line 375: The acid-catalyzed mechanism is mentioned here and even mentioned as one of the highlights of the paper, but the discussion is minimal, or inexistent here. This can be an interesting and important point, so please provide some elaboration of this topic and provide some unbiased citations of relevant literature on the topic.

Response: We have added some discussion and citations of relevant literature on this topic. We have added that "The mixture of SOA and sulfate have been detected in our chamber experiment, indicating the involvement of inorganic salts in the SOA formation. Previous studies have demonstrated the enhancement of SOA production in the presence of inorganic sulfate (Beardsley and Jang, 2015; Kuwata et al., 2015), and this is because sulfate can catalyze carbonyl heterogeneous reactions, and consequently, lead to SOA production (Jang et al., 2002; Jang et al., 2004)." (line 417-422 in the revision).

The references: Beardsley, R.L., and Jang, M.: Simulating the SOA formation of isoprene from partitioning and aerosol phase reactions in the presence of inorganics,

Atmospheric Chemistry and Physics Discussions, 15, 33121-33159, 10.5194/acpd-15-33121-2015, 2015. Kuwata, M., Liu, Y., McKinney, K., and Martin, S.T.: Physical state and acidity of inorganic sulfate can regulate the production of secondary organic material from isoprene photooxidation products, Physical chemistry chemical physics : PCCP, 17, 5670-5678, 10.1039/C4CP04942J, 2015.

28. Line 380: The large contribution of GDI emissions here is argued, but there is no data collected or model performed (even if just conceptual) to really quantify, at least semi-quantitatively, what the contribution could be with respect to other sources. The only discussion I recall is on the elemental composition regarding tracers to distinguish different sources, but no discussion to quantify their potential contribution (also in what terms? Mass? Number? Something else?). So the "considerable potential contribution" statement is a vague ill-posed guess that is not really proven here as the data are presented. One could try to estimate the contribution by calculating for example estimated total contribution of GDI engines (by multiplying the fuel-based emission factor by the fraction of fuel consumed by GDI vehicles in the area) vs. the total particle burden in the region. or some other estimate exercise of this sort. Otherwise the sentence cannot be supported by the evidence provided.

Response: In this part, what we want to say is that GDI-engine vehicles emitted a large amount of both primary and secondary organic aerosols. In order to clarify our meaning, we have removed "We highlight the considerable potential contribution of GDI-engine vehicles to both primary and secondary organic aerosols". We have added "Our results indicate that GDI-engine vehicles emitted a large amount of both primary and secondary organic aerosols. PM number emission of organic particles from GDI-engine vehicle were $2.9 \times 10^9$ particles (kg fuel)-1. Secondary organic particle was predominant in the secondary aerosols, accounting for 80-85%% particles in the chamber" (line 437-441 in the revision).

29. Lines 386-388: This is a clear example, among others, of biased representation (or lack of) previous work in this paper. There have been very many previous (definitely

previous to 2017) studies showing that individual particles, including organics come internally mixed with other particles, so this statement is not very true (especially in terms of "recent") and biased in terms of literature discussion and citations.

Response: We have removed "Recent measurements indicate that most OA exists as an internal mixture with other aerosols, and the distribution of this mixture depends on the formation mechanism of OA (Zhu et al. 2017)" in line 466-468. We have added citations and discussions of relevant articles. We have added "OA were composed of various types of chemical compounds with varying absorption properties (mixing state), which were determined by the emission sources, the formation mechanism (Zhu et al. 2017), and the source regions (Laskin et al., 2015). Primary OA from biomass burning was co-emitted with soot (black carbon), inorganic salts, and fly ash, producing internally and externally mixed particles in which the organic components were present in different relative abundance (Lack et al., 2012). Similarity, primary OA in the exhaust of gasoline and diesel vehicles were mixed with Ca, P, Mg, Zn, Fe, S, and minor Sn inorganic compounds (Liati et al. 2018). In addition, previous measurements indicated that SOA usually exists as an internal mixture with other aerosols, such as sulfate, ammonium, or nitrate (Zhu et al., 2017). Our results showed that the POA emitted from GDI-engine vehicle were mixed with soot, inorganic components such as Ca, P, and Zn. Some of the SOA formed in the smog chamber were mixed with sulfate. The complexity of mixing state makes it difficult to characterize the properties of OA" (line 447-460 in the revision). The references: Laskin, A., Laskin, J., and Nizkorodov, S.A.: Chemistry of Atmospheric Brown Carbon, Chem Rev, 115, 4335-4382, 10.1021/cr5006167, 2015. Liati, A., Schreiber, D., Arroyo Rojas Dasilva, Y., and Dimopoulos Eggenschwiler, P.: Ultrafine particle emissions from modern Gasoline and Diesel vehicles: An electron microscopic perspective, Environ Pollut, 239, 661-669, 10.1016/j.envpol.2018.04.081, 2018. Lack, D.A., Langridge, J.M., Bahreini, R., Cappa, C.D., Middlebrook, A.M., and Schwarz, J.P.: Brown carbon and internal mixing in biomass burning particles, Proceedings of the National Academy of Sciences, 109, 14802-14807, 10.1073/pnas.1206575109, 2012.

30. Lines 391-393: Again limited (poor) literature citation choice and a bit of simplistic view of the effect of internal mixing on aerosol radiation interaction (which I think is what the authors refer to here). Please improve the discussion and support it with some more reprehensive and balanced citations judiciously chosen from the large body of previous literature in the field. Another such example appears in lines 399 to 400.

Response: For the effect of internal mixing on aerosol radiation interaction, we have added citations and discussions of relevant articles. We have added that "Lang-Yona et al. (2010) found that for aerosols consisting of a strongly absorbing core coated by a non-absorbing shell, and the Mie theory prediction deviated from the measurements by up to 10%. Moreover, atmospheric aging process, involving aqueous-phase aging and atmospheric oxidation, can either enhance or reduce light absorption by OA (Bones et al., 2010). The condensation process may result in a dramatic enhancement of hydrolysis of OA compounds, affecting their absorption spectra (Lambe et al., 2013)" (line 460-466 in the revision). The references: Bones, D.L., Henricksen, D.K., Mang, S.A., Gonsior, M., Bateman, A.P., Nguyen, T.B., Cooper, W.J., and Nizkorodov, S.A.: Appearance of strong absorbers and fluorophores in limonene-O3 secondary organic aerosol due to NH4+-mediated chemical aging over long time scales, Journal of Geophysical Research, 115, 10.1029/2009JD012864, 2010. Lambe, A.T., Cappa, C.D., Massoli, P., Onasch, T.B., Forestieri, S.D., Martin, A.T., Cummings, M.J., Croasdale, D.R., Brune, W.H., and Worsnop, D.R.: Relationship between Oxidation Level and Optical Properties of Secondary Organic Aerosol, Environ Sci Technol, 12, 6349-6357, 10.1021/es401043j, 2013. Lang-Yona, N., Abo-Riziq, A., Erlick, C., Segre, E., Trainic, M., and Rudich, Y.: Interaction of internally mixed aerosols with light, Phys Chem Chem Phys, 12, 21-31, 10.1039/B913176K, 2010.

31. Lines 403 to 404: In this case, I would say the sentence is flatly untrue; there are plenty of studies on the optical properties of POA, and the authors should discuss them and cite them accordingly.

Response: We have removed "However, studies on the optical properties of OA have

been mainly focused on SOA, and only a few studies dealt with POA" in line 486-487.

We have added citations and discussions on the optical properties of POA. We have added that "Lang-Yona et al. (2010) found that for the aerosols consisting of a strongly absorbing core coated by a non-absorbing shell, and the Mie theory prediction deviated from the measurements by up to 10%. Moreover, atmospheric aging process, involving aqueous-phase aging and atmospheric oxidation, can either enhance or reduce light absorption by OA (Bones et al., 2010). The condensation may result in a dramatic enhancement of hydrolysis of OA compounds, thus affecting their absorption spectra (Lambe et al., 2013)" (line 460-466 in the revision).

In line 483-486, we have added that "These results push forward the understanding on the mixing state and chemical composition of both POA and SOA. The experimental data will benefit the parameterization of vehicles emissions in numerical models dealing with urban air pollution." The references: Bones, D.L., Henricksen, D.K., Mang, S.A., Gonsior, M., Bateman, A.P., Nguyen, T.B., Cooper, W.J., and Nizkorodov, S.A.: Appearance of strong absorbers and fluorophores in limonene-O3 secondary organic aerosol due to NH4+-mediated chemical aging over long time scales, Journal of Geophysical Research, 115, 10.1029/2009JD012864, 2010. Lambe, A.T., Cappa, C.D., Massoli, P., Onasch, T.B., Forestieri, S.D., Martin, A.T., Cummings, M.J., Croasdale, D.R., Brune, W.H., and Worsnop, D.R.: Relationship between Oxidation Level and Optical Properties of Secondary Organic Aerosol, Environ Sci Technol, 12, 6349-6357, 10.1021/es401043j, 2013. Lang-Yona, N., Abo-Riziq, A., Erlick, C., Segre, E., Trainic, M., and Rudich, Y.: Interaction of internally mixed aerosols with light, Phys Chem Chem Phys, 12, 21-31, 10.1039/B913176K, 2010.

32. Figure 2: As mentioned earlier, specify that this distribution is from the electron microscopy analysis, and overlap or add a side plot with the size distribution from the DMS500.

Response: We added the size distribution by the DMS500 instrument as Figure S2.

33. Figure 4: For regulatory applications, it could be interesting to generate a similar figure but for estimated mass (or at least volume) fractions.

Response: The mass and volume fractions emitted by the gasoline vehicles were also a concern for us. Because of the technological limitation, we couldn't test them in this experiment. We tried to discuss the mass by citing relevant publications.

In the revision, the following descriptions are added in line 353-357. "Relevant studies have also shown that the primary carbonaceous aerosols (element carbon + primary organic aerosol) accounted for 85 % of the PM in the GDI vehicles, suggesting that carbonaceous aerosols were the major contributors in the PM from GDI gasoline vehicles (Du et al., 2017)." The references: Du, Z., Hu, M., Peng, J., Zhang, W., Zheng, J., Gu, F., Qin, Y., Yang, Y., Li, M., Wu, Y., Shao, M., and Shuai, S.: Gasoline direct injection vehicles exceed port fuel injection ones in both primary aerosol emission and secondary aerosol formation, Atmospheric Chemistry and Physics Discussions, 1-31, 10.5194/acp-2017-776, 2017.
* * *
[Figure]

Figure S2. Size distribution of particles emitted from the GDI-engine gasoline vehicles by the DMS500 instrument.

**Fig. 1.** Figure S2. Size distribution of particles emitted from the GDI-engine gasoline vehicles by the DMS500 instrument

[Figure]

Figure S4. Size distribution of particles emitted from the GDI-engine gasoline
vehicles by the DMS500 instrument during the Beijing driving cycle.

**Fig. 2.** Figure S4. Size distribution of particles emitted from the GDI-engine gasoline vehicles
by the DMS500 instrument during the Beijing driving cycle

---

## Author Response (AR2)

**Point-to-point responses to the comments:**

The authors addressed most of my concerns. However, I think there are still a few small issues:

*Response:* Many thanks to the reviewer for the comments and suggestions. Here I describe the point-to-point responses to the comments and questions.

1.  Verb tense consistency in the paper is still sometimes an issue.

*Response*: The past tense is used in the descriptions for the experiments, and the present tense is used in the general narration. We have carefully checked and revised the full text again.

2.  More importantly, though, I am still a bit confused by the statements about the PM emissions being higher for hot start state with respect to cold start state, that's a bit counterintuitive for me. It would be good to comment on some possible explanations for these statements/findings. For example, at line 471 the authors mention: "The PM emission was the highest under the hot stabilized running state, followed by those under the hot start, cold start" (this is mentioned also in other parts of the paper) but looking at figures 3 and 4, it seems to me that during the cold start state the vehicle emitted substantially more particles than during the hot start state. Am I misinterpreting the sentences, or am I misreading the figures, or is there a contradiction? Anyway, it would be good to clarify this and to provide a potential explanation.

*Response*: Many thanks for pointing out the misleading description. Our results showed that the total PM emissions under hot start state were slightly higher than that of under the cold start state. The higher emissions under hot start state may be ascribed to the conducting time of the vehicle engine. The hot start test was conducted within 5 mins after the cold start test. PM emissions from GDI vehicles are relatively less affected by ambient temperature for engines warmed up for 30 min (Cotte et al., 2001). Therefore, the emissions under the hot start could be higher than, actually close to, that under the cold start state.

The particles shown in Figures 3 and 4 were only those in the size range of accumulation mode. Although the total PM emission were higher under hot start state than that under the cold start state, the comparison of particles in the size range of accumulation mode indicated that the emissions for particles in this size mode were higher under the cold start state than under the hot start sate (Figures 3, 4). Under the cold start state, larger proportions of particles were present in the size range of accumulation mode particles. This can be attributed to the less efficiency of the vaporization of fuel droplets in the combustion cylinder under the cold start state (Chen et al., 2017).

In the revision, we added some explanations for these statements/findings in lines 296-308: "The higher emission of particle in term of number for the GDI vehicle under the hot start state can be ascribed to the experimental time of the vehicle engine. The hot start test in this study was conducted within 5 mins after the cold start test. The PM emissions from GDI vehicles were relatively less affected by ambient temperature for the initial 30 minutes during the warming up of the engines (Cotte et al., 2001). This may lead to the high value of the PM emission for the hot start state which is slightly higher than that for the cold start state. Although the total PM emission were higher under hot start state than that under the cold start state, the comparison of those in the size range of accumulation mode indicates that the particulate emissions for this mode of particles were higher under the cold start state than under the hot start sate (Fig. 3). This can be attributed to the less efficiency of the vaporization of fuel droplets in the combustion cylinder under the cold start state (Chen et al., 2017)."

[revised manuscript text omitted]